# Mineral Surface-Templated Self-Assembling Systems: Case Studies from Nanoscience and Surface Science towards Origins of Life Research

**DOI:** 10.3390/life8020010

**Published:** 2018-05-08

**Authors:** Richard J. Gillams, Tony Z. Jia

**Affiliations:** 1Earth-Life Science Institute, Tokyo Institute of Technology, 2-12-1-IE-1 Ookayama, Meguro-ku, Tokyo 152-8550, Japan; ric.gillams@gmail.com; 2Structural Genomics Consortium, Nuffield Department of Medicine, University of Oxford, Old Road Campus Research Building, Oxford OX3 7DQ, UK

**Keywords:** origins of life, prebiotic chemistry, self-assembly, nanoscience, surface science

## Abstract

An increasing body of evidence relates the wide range of benefits mineral surfaces offer for the development of early living systems, including adsorption of small molecules from the aqueous phase, formation of monomeric subunits and their subsequent polymerization, and supramolecular assembly of biopolymers and other biomolecules. Each of these processes was likely a necessary stage in the emergence of life on Earth. Here, we compile evidence that templating and enhancement of prebiotically-relevant self-assembling systems by mineral surfaces offers a route to increased structural, functional, and/or chemical complexity. This increase in complexity could have been achieved by early living systems before the advent of evolvable systems and would not have required the generally energetically unfavorable formation of covalent bonds such as phosphodiester or peptide bonds. In this review we will focus on various case studies of prebiotically-relevant mineral-templated self-assembling systems, including supramolecular assemblies of peptides and nucleic acids, from nanoscience and surface science. These fields contain valuable information that is not yet fully being utilized by the origins of life and astrobiology research communities. Some of the self-assemblies that we present can promote the formation of new mineral surfaces, similar to biomineralization, which can then catalyze more essential prebiotic reactions; this could have resulted in a symbiotic feedback loop by which geology and primitive pre-living systems were closely linked to one another even before life’s origin. We hope that the ideas presented herein will seed some interesting discussions and new collaborations between nanoscience/surface science researchers and origins of life/astrobiology researchers.

## 1. Introduction

What fundamental requirements does Earth or any planet need to produce and sustain life? This is a fundamental question related to our own genesis and history, and the study of origins of life seeks answers and plausible explanations to this and related questions regarding how life on Earth came into existence and how life could exist or arise elsewhere in the universe. The exploration of space is a natural extension of this line of inquiry and provides an array of technical challenges that has driven innovation and continues to inspire scientists to understand their environment, both on this planet and beyond. This overarching aim disguises an inventory of scientific understanding that bridges a broad range of disciplines from astrophysics to microbiology, from chemistry to geology, and requires collaborative efforts where the boundaries of research fields become blurred or new fields must be created. Specifically, there are many research fields which lie at the periphery of origins of life research, including nanoscience and surface science, that are willing and able to contribute to origins of life research, yet much of the research remains unknown to those in the field. How do we engage and collaborate with such research communities in order to incorporate their ideas and answer fundamental questions related to life’s origins?

Recent studies in origins of life research have focused on how the basic building blocks of life (i.e., amino acids, nucleotides, lipids, etc.) could have been formed on early Earth or on other planets. As the origin of life on Earth likely involved aqueous chemistry [1], much prebiotic chemistry research has focused on this. Yet interfacial and supramolecular chemistry should also be considered in a prebiotic context, as these systems result in phenomena that can catalyze reactions and facilitate novel reaction pathways inaccessible in solution [2]. This increased structural, functional, and/or chemical complexity is achieved through, for example, increased local concentrations of adsorbed reactants (pushing the equilibrium towards product formation) [3,4], orientational specificity of adsorbed reactants (which would result in a preferred face or moiety of an adsorbed molecule to be accessible (or inaccessible) to catalysts in solution) resulting in catalytic selectivity [5,6], or through the provision or accession of electrochemical driving forces such as in mineral semiconductors [7,8]. The most common interfaces available on primitive Earth or other extraterrestrial bodies are mineral surfaces. Minerals are produced during the formation of terrestrial planets [9], and various minerals, including apatites (Ca_5_(PO_4_)_3_(F,Cl,OH)) [10], phosphate minerals such as schreibersite ((Fe,Ni)3P) [11,12], and borates such as colemanite (Ca_2_B_6_O_11_·5H_2_O) [13,14] (although boron is fairly common in different sedimentary rocks, a mechanism by which high concentrations of specific borate minerals could have accumulated on early Earth is unclear [15]), can catalyze the synthesis of nucleotides and amino acids from simple chemical precursors like phosphates, nucleosides, or formamide. These nucleotides and amino acids can adsorb onto clay minerals such as montmorillonite ((Na,Ca)_0.33_(Al,Mg)_2_(Si_4_O_10_)(OH)_2_·nH_2_O), illite ((K,H_3_O)(Al,Mg,Fe)_2_(Si,Al)_4_O_10_[(OH)_2_,(H_2_O)]), and kaolinite (Al_2_Si_2_O_5_(OH)_4_) [16,17,18], which can catalyze their polymerization into more complex biopolymers such as nucleic acids [19,20,21,22] and peptides [23,24,25,26,27]. Various mineral surfaces such as montmorillonite [28], silicates (such as aluminum silicate, Al_2_SiO_5_), carbonates (such as hydrotalcite, Mg_6_Al_2_CO_3_(OH)_16_·4(H_2_O)) [29], and sulfides (such as pyrite, FeS_2_), among others [30] have also been shown to promote fatty acid vesicle formation, an important process in the initial development of primitive cells by which biopolymers become stably encapsulated within a compartment, i.e., a protocell. Thus, mineral–water interfaces are clearly important for prebiotic chemistries leading to primitive life both on Earth [31] as well as across the entire cosmos [32]. As mineral surfaces are likely to be similar in structure universally, understanding the catalytic properties of terrestrial mineral surfaces in the laboratory would give us more information on how life could have emerged on other rocky planets.

In this review, we examine how mineral surfaces can template and promote prebiotic chemical and biomolecular self-assemblies that were likely an essential stage in the origin of life on Earth by presenting case studies from nanoscience and surface science, such as organization of small organic monolayers on mineral surfaces, secondary structure changes of nucleic acids upon mineral surface binding, assembly of peptide amyloids by mineral surfaces, and mineralization of new mineral surfaces by self-assembled chemical systems. Self-assemblies such as folded proteins or DNA double helices [33] are important in extant life as they provide structures and functions that non-assembled systems are simply unable to provide. Similar assemblies may have been especially important for early living systems long before the emergence of genetic evolution due to their ability to achieve higher complexity compared to simpler structural systems. Self-assembly is also often spontaneous and energetically favorable, taking advantage of non-covalent bonds (covalent bonds such as phosphodiester or peptide bonds in biopolymers produced by condensation were generally thermodynamically unfavorable on early Earth [34]), including H-bonding, aromatic stacking, electrostatics, and hydrophobic interactions [35,36,37] and can be perturbed through simple processes available on early Earth such as heating [38], hydration/drying [39], pH changes [40], and magnetic [41] and electric fields [42], allowing for dynamic assembly and disassembly processes. We present some case studies from surface science and nanotechnology that may be unfamiliar to the origins of life/astrobiology research communities to highlight the breadth of assembly-promoting functions that different mineral surfaces may provide, and the increase in structural, functional, and/or chemical complexity arising from these self-assemblies. It is clear from these examples that the origins of life research field can be enhanced through closer collaboration with colleagues in the nanoscience and surface science research fields.

## 2. Case Studies

### 2.1. Simple Organic Molecules

In addition to adsorption of biomonomers such as nucleotides and amino acids, mineral surfaces are able to adsorb or interact with other various small organics including small carboxylic acids like salicylic acid onto kaolinite (Al_2_Si_2_O_5_(OH)_4_) and aluminum oxide (Al_2_O_3_) [43] and simple alcohols such as methanol or isopropanol onto calcite (CaCO_3_) [44]. The adsorption of these small organic molecules is often governed by the functional groups on a given molecule, the structure and charge of the mineral surface, and also the solvent [43], often generating mono- or poly-layers of the small organic molecule on the mineral surface [44,45,46]. Ordered surface organic layers (in this case, stearic acid) have even been studied in the context of calcite crystal nucleation [47].

Stipp and coworkers studied the microstructure of ethanol layers adsorbed to the surface of calcite after incubation in liquid ethanol or water/ethanol mixtures and determined that ethanol monolayers form readily [45,48,49]. In the case of incubation with pure ethanol, two organic layers formed, with the layer closest to the calcite surface having a thickness equal to the molecular length of a single ethanol molecule [45] (Figure 1). These ethanol molecules adsorb perpendicular to the mineral surface in a highly ordered manner, while the second ethanol layer is much thicker, slightly less ordered, and is separated from the first layer by a gap. All of the polar hydroxyl groups in the first ethanol layer are oriented towards the ionic calcite surface, while the non-polar ends of the ethanol are oriented away from the mineral surface; this was also observed in studies with other alcohols like pentanol and propanol [44]. If the upper layer is removed, then the assembly of single alcohol monolayers onto calcite or any other mineral would change the surface to be hydrophobic. Where initially more hydrophilic groups would interact preferentially with and be adsorbed directly by the mineral surface, after a mineral surface is treated with small organic amphiphiles, this new hydrophobic surface could then restrict the ability of water or hydrophilic molecules such as metal ions from accessing the mineral surface [48].

One may also imagine a scenario where only part of a surface is populated by an organic monolayer, such as on a heterogeneous rock surface with multiple mineral types that have different adsorption propensities of organics such as shales, which are sedimentary rocks that contain multiple different mineral species [50,51]. Consider a shale which has been eroded slightly by weathering such that the exposed sections contain the surfaces of multiple minerals including clays, quartzes, carbonates, calcites, and others. Amino acids have been shown to have specific preferences for binding to certain minerals, such as aspartic acid preferentially adsorbing onto calcite (CaCO_3_) and glycine, serine, isoleucine, leucine and arginine preferentially adsorbing onto quartz (SiO_2_) [52]. Different clays have also shown different propensities for adsorption of various simple organics, including small polar organics like formamide or urea strongly adsorbing onto kaolinite (Al_2_Si_2_O_5_(OH)_4_) and various aromatic compounds like benzene and phenol interacting with Cu-Montmorillonite (copper (II) with (Na,Ca)_0.33_(Al,Mg)_2_(Si_4_O_10_)(OH)_2_·nH_2_O) [53]. A diverse heterogeneous organic microenvironment could form on such shale surfaces, providing localized reactions on each specific mineral. This microenvironment could have also facilitated interactions or reactions between the different heterogeneous organic pockets due to the adjacent proximity of these reactions. If so, such heterogeneous environments could assist early chemical systems in diversifying and incorporation or coupling of different types of reactions together to form larger, more complex chemical networks.

### 2.2. Nucleic Acids

Nucleic acids are the genetic materials for all known modern life, and thus it is important to understand how these materials emerged, evolved, and what governs their function [33]. The structures assembled by nucleic acids are not merely one-dimensional primary sequence strings of nitrogenous bases, but rather are due to complex folding into secondary, tertiary, and quaternary structures that result in geometries and assemblies essential for cellular function [54]. Within a cell, DNA molecules are typically assembled into helices, which allow for complementary strands to hybridize. Major and minor grooves are generated in the phosphate backbones of these helices that allow for binding of specific proteins [55]. With the help of certain enzymes, these assemblies temporarily come apart to allow access to the bases during important cellular processes including transcription [33]. Specific, complex assemblies are also required for the function of various ribozymes, primitive RNA-based catalysts that may have been important during the initial emergence and evolution of primitive genetic systems through their potential catalysis of RNA polymerization [56,57]. Although nucleotides can oligomerize somewhat under prebiotically plausible conditions, for example through high-energy chemical activation in solution [58] (in the absence of an activated group, nucleotides have so far not been shown to be able to polymerize at all) or on clay mineral surfaces [19], there is still no known mechanism by which nucleotides can efficiently polymerize into nucleic acid polymers and subsequently replicate and evolve in the absence of enzymes [59]. Despite this limitation in knowledge, there has still been plenty of work probing nucleic acid adsorption onto mineral surfaces. The assembly of nucleotide monomers adsorbed onto mineral surfaces into monolayers has been postulated by some as an important process that promoted interactions between nucleotides and other important biomolecules including amino acids and carbohydrates [60]. Both single- and double-stranded nucleic acids have also been shown to adsorb onto various mineral surfaces, including mica (general) [61], clay minerals [62], calcites, carbonates, and metal oxides [63].

Typically, double-stranded DNA preferentially assembles into a B-form helical conformation in standard aqueous physiological conditions [64]. However, upon binding to the clay mineral kaolinite (Al_2_Si_2_O_5_(OH)_4_), DNA changes to a Z-form helical conformation [62], a unique left-handed helical form apparent (though rare) in nature [65]. Although some functions for Z-form DNA have been discovered, including increasing of mutagenesis and deletion rate near Z-form DNA loci [66,67], no specific biological function has yet been determined other than their apparent role in the function of certain poxviruses [68]. Binding to montmorillonite ((Na,Ca)_0.33_(Al,Mg)_2_(Si_4_O_10_)(OH)_2_·nH_2_O), however, does not change the helical conformation at all [62]. In an early Earth scenario, it is possible that binding to clay mineral surfaces such as kaolinite (Al_2_Si_2_O_5_(OH)_4_) provided access to novel helical structures for nucleic acids, which normally are only possible in non-standard conditions, for example A-form transitions which normally require high relative humidity or Z-form transitions which normally require high salt concentrations [68,69].

Although it is clear, due to its dual role as a genetic information carrier/proliferator and as a catalyst, that RNA would have likely been more important in the initial emergence of life than DNA, there is a current lack of research on effects of mineral surface adsorption on RNA conformations, save for some computational studies [70]. However, given the fact that, similar to DNA, RNA can also transition between various helical forms depending on environmental factors [71], future studies of mineral surface-effected changes in adsorbed RNA conformations would be useful and of interest to the origins of life research community. For example, normally nucleotides within a standard A-form helical RNA exists in the C3′-endo *anti* sugar pucker conformation, while Z-form RNA is composed of repeating nucleotides with alternating C2′-endo *anti* and C3′-endo *syn* sugar puckers, often with alternating cytidine-guanosine bases [72]. However, some small functional RNAs such as riboswitches or aptamers contain Z-like steps, which are localized areas of a nucleic acid strand generated when two adjacent nucleotides exist in the *anti* (5′) and *syn* (3′) conformation [73]. Z-form RNA perhaps composed of cytidine-guanine repeats (which may have been more abundant than adenine and uridine on the primitive earth as cytidine and guanine are less error prone and polymerize orders of magnitude faster in a nonenzymatic replicating RNA system than adenine and uridine [74]), RNA containing Z-like steps, or any other novel RNA structures that could have been induced in a primitive replicating genetic system, perhaps by mineral adsorption, could have potentially allowed primitive nucleic acid systems to explore additional structure space during evolution in a search for novel or more efficient functions or catalytic activity, respectively.

Additionally, DNA molecules have been shown to adsorb to mica (general) and form specific structures under different ionic conditions. Salts like magnesium chloride and cobalt chloride supply divalent cations, which bridge the negatively charged mica surface with the negatively charged nucleic acid phosphate backbone [75], resulting in DNA adsorbing onto a mica surface; a salt like sodium chloride, which only provides a monovalent cation, does not promote DNA adsorption at all [61]. Different divalent counterions can result in different structural moieties of DNA upon mica surface binding: magnesium cations resulted in condensed and coiled DNA (with some topography) on the surface, while cobalt cations induced formation of flat two-dimensional monolayers. This is due to the fact that cobalt cations have a greater affinity for mica than magnesium cations [76], resulting in tighter binding between the DNA and the mica surface. This binding could lead to a monolayer structure where most of the DNA is adhered to the surface, as opposed to magnesium-induced coiled DNA structures where much less of the DNA is adhered (but it is still immobilized) to the surface. In addition to cobalt cations, which could have been introduced (although perhaps not in high enough concentration to effect the structural changes described above) into prebiotic aqueous systems through weathering of cobalt-containing minerals [77], other more abundant prebiotic divalent cations that show a strong ability to promote DNA adsorption on mica include nickel and zinc. The ionic radii of these two divalent cations (similar to cobalt and magnesium; although the ionic radii of cobalt and magnesium cations are similar, magnesium cations do not contain any *d* electrons and are thus unable to form as many complexes as cobalt, explaining why cobalt has a greater affinity for mica than magnesium) are small enough to fit into mica cavities, resulting in a stronger adhesion of DNA to the mica surface [76]. Ferrous iron could have also been a strong promoter of DNA-mica surface adsorption on early Earth, as its ionic radius is similar to that of cobalt and zinc cations [78]. It would be of great interest to study the structure of adsorbed DNA and RNA on mica surfaces in the presence of these additional divalent cations rather than cobalt due to their higher concentration than cobalt on the early Earth (in particular ferrous iron, which was very abundant in early Earth oceans [79]) to determine whether mica surfaces coupled with prebiotically available divalent cation solutions could have contributed to additional structural order and complexity for primitive nucleic acids.

The structuring of nucleic acids, in particular RNA, onto mineral “scaffolds” in different ionic conditions could have also provided an anchoring surface for easy and rapid access to a complex library of different sequences that could be probed for catalytic activity or binding through flowing substrates. Such an arrayed nucleic acid system would be structurally similar to DNA nano-arrays assembled on mica (general) surfaces that have been developed by materials scientists [80] (Figure 2). This idea could also be extended to include the possibility of assembly on mineral surfaces of primitive nucleic acid origamis [81], modern synthetically-produced structures that rely on specifically-designed base–base interactions to result in assembly into specific shapes. Although much nucleic acid origami research has focused only on the novel structural aspects within DNA-only systems, recently researchers have made designed functional DNA origamis, including nanocages for encapsulation of small molecules in drug delivery [82,83] or lipid membrane extensions that promote cell–cell adhesion [84]. In addition to DNA origamis, more relevant to early Earth systems, RNA origamis have been developed [85,86,87]. One such system involved using modified versions of the Tetrahymena GI ribozyme [88] and adding short, specifically-designed recognition sites, i.e., short runs of a few bases, to some regions of the modified ribozyme. These modules were then allowed to assemble, and due to the designed interactions between certain recognition sites, modular trimers and tetramers assembled into triangular and square-like structures, respectively. Although the structure is by itself already novel, both the assembled triangle and square structures resulted in greater catalytic activity than just its separate components [86]. This demonstration of structural assemblies of catalytic RNA resulting in greater catalytic activity suggests that in a primitive genetic system on the early Earth, modular self-assemblies combining multiple short catalytic RNA could have also resulted in catalytic enhancements (or even novel functions). However, one must keep in mind that the examples provided from the nucleic acid origami fields all require very specific base-designs, a feature that an early Earth system would not have had access to. Nevertheless, at least in [86], each of the recognition sites contained only 4–8 nucleotides, suggesting that even in a primitive system, the required recognition site sequences are still reasonably short (such that there is a possibility that they could have formed through random sequence polymerization, especially if there was a much greater abundance of cytidines and guanines in the available nucleotide pool for the same reasons as described above [74]) and thus it still could have been possible for simpler, shorter self-assembled modular catalytic structures to have formed.

### 2.3. Peptides

One of the most important findings in the field of degenerative disease medicine (and also, as it turns out, peptide nanoscience) was the discovery that a short peptide of around 40 residues, amyloid beta, assembles into filamentous plaques in the brain of Alzheimer’s disease patients [89]. This physiologically produced peptide, which misfolds and aggregates into oligomers, is generally the cause of this deadly neurodegenerative disease [90] likely through inhibition of certain tissue functions, although it is not well understood and other hypotheses exist [91,92]. Since this discovery, many other proteins and short peptides have been implicated in diseases arising from the assembly of amyloid fibrils [93], and at the same time these discoveries have inspired much new research in small peptide self-assemblies. In an attempt to develop a nanotubular material relevant to nanowire nanofabrication, Gazit and coworkers discovered that within the amyloid beta peptide, a short diphenylalanine dimer was able to self-assemble into distinct nanotubes, giving the first demonstration that even peptides with as little as two subunits can self-assemble into robust nanostructures [94]. Since then, many other small, self-assembling peptides have been discovered through experimental and computational methods [95,96]. These assemblies contain varied architectures including fibers, films, and hydrogels [97], and have a wide range of engineering applications including as antibacterial agents [98], semiconductors [99], and much more [100].

Although these examples are all of modern peptide self-assemblies in applicational settings, peptide self-assemblies, including those templated by mineral surfaces, are still relevant to prebiotic chemistry as they are structures that can provide structural and functional complexity to early living systems. Although amino acids produced on early Earth through atmospheric discharge [101,102], from extraterrestrial sources including meteorites [103] and in undersea hydrothermal systems [104,105] can then polymerize into simple peptides through volcanic-gas activation of amino acid monomers [106,107] or on clay mineral surfaces coupled with wet–dry cycles [23,24,25,108], the ability of mineral surfaces to catalyze the polymerization of very long peptides with catalytic functions remains poorly studied and understood. Thus, self-assembly of short peptides may be one energetically favorable way for an early Earth system to access emergent architectures or functional properties. As mineral surfaces present sites capable of interaction with the various functional groups on peptides, these surfaces could have been anchors for promoting short peptide self-assembly.

Many studies in the nanoscience field have probed the role of mineral surfaces in self-assembling peptide systems. For example, 16-mers of polyglycine can aggregate into ordered helical and β-sheet secondary structures on oxide surfaces including anatase (TiO_2_) and amorphous SiO_2_ [109]. This demonstration is important as secondary structures (as well as tertiary and quaternary structures) are necessary for proteins to achieve catalytic function [69], suggesting that mineral surfaces could have assisted in developing primitive peptide function by promoting the formation of more complex structures not possible in solution, similar to what was observed above in RNA upon kaolinite (Al_2_Si_2_O_5_(OH)_4_) binding. The assembly of other peptides, such as longer amyloids (around 100 residues) or short amyloid-derived peptides (eight to nine residues), into filaments or fibrils can also be templated by mica (general and muscovite, (KAl_2_(Si_3_Al)O_10_(OH)_2_)) surfaces [110,111,112] (Figure 3). On primitive Earth, these filamentous nanostructures could have aided in increasing the structural or functional complexity of early chemical systems, for example by scaffolding and/or catalyzing the polymerization of biopolymers like other peptides [113]. Nanofibers also could have been more efficient, compared to mineral surfaces, at promoting the self-assembly of additional peptide nanostructures [114]. Finally, the physical appearance of self-assembled peptide nanofibers, such as in Figure 3, appears to be fibers branching out from a central nucleation site. This bears a striking physical similarity to peptide dendrimers, which are branched polymers containing a core from which various functional peptide groups emanate [115]. Although the branching of peptide nanofibers takes place at the supramolecular scale, as opposed to the branching of peptide dendrimers taking place on the molecular scale, given the similarity in overall dendrimer-like structure perhaps branched peptide nanofibers on early Earth could have had similar functions as peptide dendrimers. For example, if peptide nanofibers contained many catalytic residues like serine, histidine, or aspartic acid [116], perhaps the supramolecular ordering afforded by the fibrils could allow the formation of primitive catalytic triads [117], just like those found in some peptide dendrimer enzymes [118].

In addition to amyloids, molecules like peptide amphiphiles [119] or amphiphilic peptides [120] are also able to assemble into nanostructures on mineral surfaces such as mica (general). Although these are the only two examples given of mineral-templated nanofiber structures, there is a wide variety of other structures and functions that self-assembled peptide amphiphiles [121] and amphiphilic peptides [122] are able to achieve in solution, each of which would be an interesting case study for origins of life researchers in the context of mineral-surfaces. For example, simple amphiphilic peptides like A_6_D or A_6_K (A: alanine, D: aspartic acid, K: lysine) have surfactant-like properties and can form structures such as nanotubes and nanovesicles [123,124]. In particular, surfactant-like peptide nanovesicles, which are about 50 nm in diameter and similar in size to oleic acid vesicles used in many origins of life studies [125], could have been an encapsulation alternative to oleic acid vesicles, which can be unstable depending on the ionic conditions (e.g., in high-magnesium ion conditions and without a magnesium ion chelator like citrate present [126]). Peptide amphiphiles (although it is unclear exactly how a long aliphatic tail could have attached itself onto a peptide), on the other hand, can form nanofibers which have the ability to affect local geology by promoting mineralization of new hydroxyapatite (Ca_5_(PO_4_)_3_(OH)) mineral surfaces [127].

Given the appropriate conditions and peptides (for example, short tripeptides like lysine-tyrosine-phenylalanine, i.e., KYF, [95]) a dense network of peptide nanofilaments assembled on a mineral surface in a confined space on the early earth, such as in mineral pores [128], could have formed a peptide hydrogel [129] with the potential to incorporate mineral particles like amorphous calcium carbonate [130]. A primitive hydrogel structure could have decreased diffusion rates [131] or slowed down the degradation of important and/or labile prebiotic nutrients [132] including amino acids or other small organics. Diffusion rate decreases in particular are very relevant to primitive RNA replication systems, as inhibition of complementary RNA strand re-annealing in viscous solutions (through slowing down RNA diffusion) is one of the only known ways by which RNA systems can overcome the “strand inhibition” problem and be able to undergo sustained replication [133]. These self-assembled peptide gels could have then also stabilized or localized primitive cells through their adhesive properties [134] or even directly affected local geology by driving the formation of hydroxyapatite (Ca_5_(PO_4_)_3_(OH)) [135] (similar to what has been seen in peptide amphiphile nanofibers) or other minerals. There are many other examples of self-assembled peptide structures independent of mineral surfaces composed of short, prebiotically available peptides with emergent properties important for the initial emergence of life including binding [136], segregation [137,138], encapsulation [139], or even catalysis [140,141]; these structures are worth further study in the context of co-assembly with mineral surfaces to determine whether additional unique structures or functions could have emerged.

### 2.4. Biomineralization

While much has been discussed about minerals promoting and templating biomolecular self-assemblies, many of which are relevant to the origin of life on early Earth, converse processes of self-assemblies promoting mineral formation also exist. These processes and reactions are important in the context of biomineralization, an important biological process by which organisms produce minerals for their own use through biocatalysis [142]. Some common biogenic minerals include structures like teeth, bones, shells, and pearls [143,144], as well as magnetite (Fe_3_O_4_) [145], coral [146], and otoliths (“ear stones”) [147]. What is very clear from the immense number of examples of biomineralization is that living organisms on Earth have the ability to dramatically alter Earth’s geology. Given that microbes [148] and viruses [149] (both of which were amongst the earliest residents on Earth) have the ability to affect local mineral compositions and that certain microbial biomineralization processes like magnetite formation potentially evolved early in Earth’s history [150], Earth’s geology has likely been affected directly by its inhabitants since (or even before) the origin of life. Previously, we mentioned that self-assemblies of specific peptides [135] and peptide amphiphiles [127] are able to effect the growth of hydroxyapatite (Ca_5_(PO_4_)_3_(OH)). Another major class of biopolymers that can also template mineral growth is polysaccharides (although there is not yet a reasonable prebiotic pathway to efficient polysaccharide synthesis), which are important in biological assemblies such as cellulose as structures in plants [151] or glycogen energy stores in animals [152]. Polysaccharides such as chitosan, normally derived from chitin in crustacean shells [153], are not only able to adsorb onto mineral surfaces like mica [154], but also self-assemble into a wide variety of nanostructures, e.g., fibers or films, on these surfaces. These nanostructures can actually act as supramolecular scaffolds or templates for the production of minerals such as hematite (Fe_2_O_3_) [155], one of the minerals found on Mars [156]. In addition to these examples, there is a large list of other self-assembled structures that promote mineralization and mineral growth that would be interesting to study in the context of the emergence of primitive “bio-” or “pre-bio-” mineralization on the early Earth [157,158,159,160,161,162] (Figure 4).

## 3. Prospective

### 3.1. Synergistic Cyclical Model of Mineral-Templated Self-Assembling Systems Promoting Mineralization

We envision a model by which mineral surfaces on primitive Earth catalyzed the synthesis and/or adsorption of simple biomolecules (such as nucleotides [10]), polymerization of monomers into polymers (such as peptides [23] or nucleic acids [19]), and eventual self-assembly (such as in peptide amyloid structures [110,112]) of these molecules into supramolecular structures (Figure 5). As we have shown in this review, these self-assemblies can afford a primitive system novel structures or catalytic activities such as DNA changing to a Z-form helical conformation upon kaolinite (Al_2_Si_2_O_5_(OH)_4_) binding [62] or amyloid self-assemblies catalyzing the synthesis of short peptides [113]. Some of the supramolecular structures assembled on mineral surfaces could have then, in turn, affected the local mineral composition through the catalysis of new mineral formation (for example, peptide hydrogels promoting hydroxyapatite formation (Ca_5_(PO_4_)_3_(OH) [135] (Figure 5), which then would have affected primitive chemistry in a circular autocatalytic fashion. It is very clear that the geology of our planet was affected directly by various non-living chemical processes while the Earth-life system was moving beyond primitive chemistries and into a living system early in its history, resulting in an interactive and symbiotic feedback loop in which geology and early pre-living systems were closely linked to one another even before life’s origin.

### 3.2. Incorporating Recent Discoveries in Nanoscience and Surface Science into Origins of Life Research

The systems presented in this review only offer a small glimpse into the realm of recent discoveries within nanoscience and surface science. While these examples are heavily application focused, the analytical tools used, mechanisms elucidated, and systems studied in these fields are still able to contribute to basic science research, similar to how basic science is able to do the same for applicational engineering research. Origins of life research is inherently an interdisciplinary field, and often involves researchers in chemistry, biology, geology, planetary science, physics, complex systems, and many other fields. Nanoscience and surface science research both certainly have a place in such an interdisciplinary field as well, as evidenced by the various examples that we have presented. We hope that through these various case studies of mineral-templated self-assemblies, researchers in the origins of life and astrobiology fields see the merit of incorporating ideas from these fields into their own research, specifically in order to study questions on how complementary mineral–chemical systems can interact, co-assemble, and co-evolve over time. However, there are other relevant prebiotic systems in which ideas from nanoscience or other materials science- or engineering-related fields would be able to contribute to, and thus we will not limit the scope of interaction between these fields just to mineral surfaces. As evidenced by the 2018 Gordon Research Conference on the Origins of Life, there has been a steady recent increase in participation in our research field by researchers from traditionally only peripherally origins of life-related fields, including nanoscientists and engineers, which is highly encouraging. This is because only through a truly interdisciplinary effort can we come to understand more about both our own origins and whether life could have possibly emerged elsewhere in the universe.

## Figures and Tables

**Figure 1 life-08-00010-f001:**
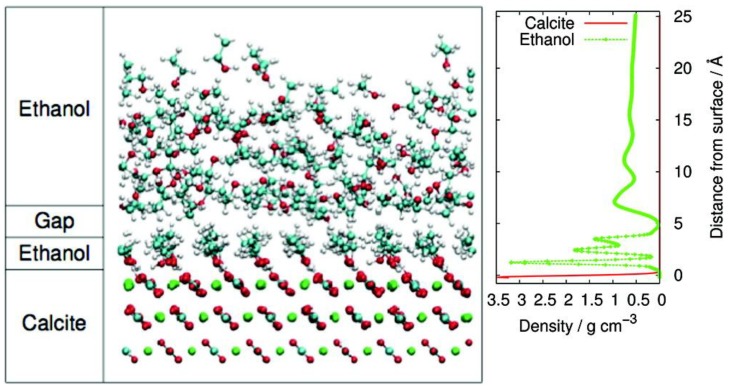
The snapshot time step from molecular dynamics simulations of ethanol layer assembly (into two distinct layers: a thin, ordered monolayer with the thickness of exactly one ethanol molecule, and a second larger less-ordered layer above the monolayer, separated by a gap) on a calcite surface, along with the ethanol spatial density profile associated with it. Reprinted with permission from Pasarín, I. S., Yang, M., Bovet, N., Glyvradal, M., Nielsen, M. M., Bohr, J., Feidenhans’l, R., and Stipp, S. L. S. 2012. “Molecular Ordering of Ethanol at the Calcite Surface.” *Langmuir* 28 (5):2545–50. [45] Copyright 2011 American Chemical Society.

**Figure 2 life-08-00010-f002:**
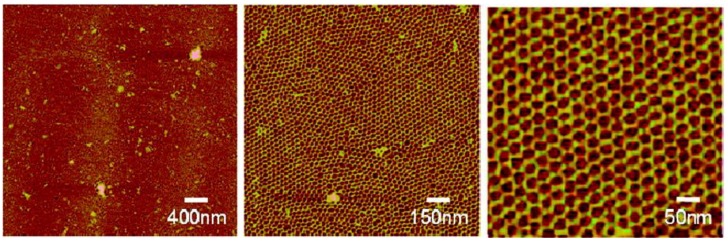
Atomic force microscope images of a structured DNA microarray self-assembled on a mica (general) surface. Reprinted with permission from Sun, X., Ko, S. H., Zhang, C., Ribbe, A. E., and Mao, C., 2009. “Surface-Mediated DNA Self-Assembly.” *J. Am. Chem. Soc.* 131 (37):13248-9. [80] Copyright 2009 American Chemical Society.

**Figure 3 life-08-00010-f003:**
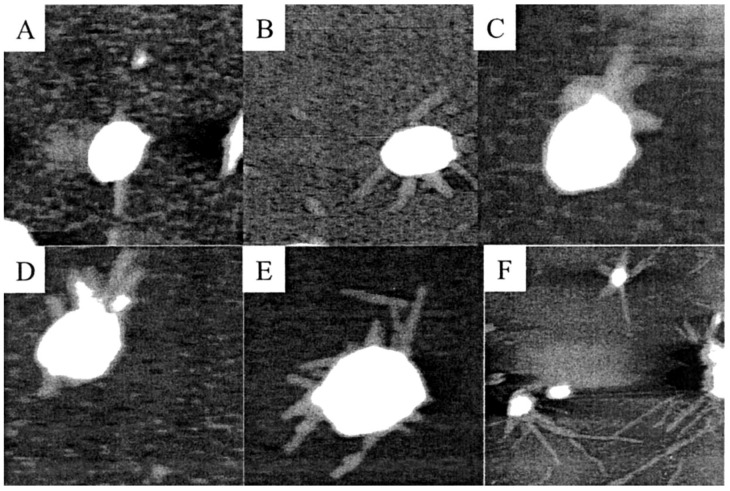
Atomic force microscope images of the assembly dynamics of SMA (an amyloidogenic light chain variable domain peptide with 114 residues) nanofibrils on a mica surface over time (**A**–**E**) show 12–16 h of incubation with image size of 1 µm × 1 µm, while (**F**) shows 6 days of incubation with image size of 5 µm × 5 µm). It appears that longer incubation times result in longer nanofibrils, such as in (**F**), as compared to shorter incubation times, such as in (**A**–**E**). Reprinted with permission from Zhu, M., Souillac, P. O., Ionescu-Zanetti, C., Carter, S. A., and Fink, A. L. 2002. “Surface-catalyzed Amyloid Fibril Formation.” *J. Biol. Chem.* 277(52):50914-22. [112] Copyright 2002 American Society for Biochemistry and Molecular Biology.

**Figure 4 life-08-00010-f004:**
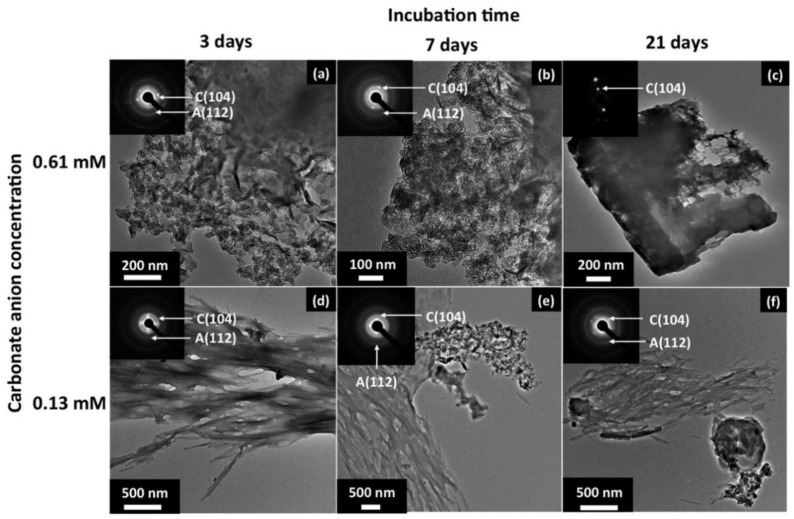
Transmission electron microscope images of calcium carbonate minerals at different nutrient concentrations and incubation times grown on a self-assembled short peptide nanostructure (Ac-VHVEVS-CONH2), suggesting that mineral growth can be effected by simple peptide self-assemblies. Reprinted with permission from Murai, K., Kinoshita, T., Nagata, K., and Higuchi, M. 2016. “Mineralization of Calcium Carbonate on Multifunctional Peptide Assembly Acting as Mineral Source Supplier and Template.” *Langmuir* 32 (36):9351–59. [157] Copyright 2016 American Chemical Society.

**Figure 5 life-08-00010-f005:**
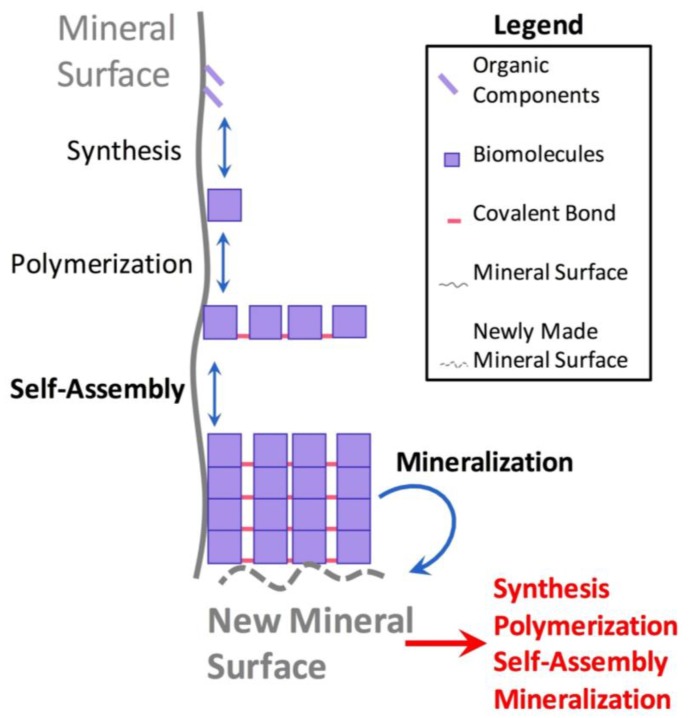
Synergistic cyclical model of mineral-templated self-assembling systems promoting mineralization. Simple precursors of biomolecules adsorbed onto a mineral surface react and produce biomolecules such as nucleotides or amino acids. Then, some of these adsorbed molecules polymerize into polymers (such as peptides or nucleic acids). Given the optimal conditions and mineral surfaces, biopolymers produced then self-assemble into supramolecular structures such as amyloid fibers or primitive nucleic acid “origamis”. Although not shown in the figure, at any stage in the adsorption, polymerization, and self-assembly pathway (which are all reversible reactions), each molecule can also desorb off of the mineral surface. However, certain self-assembled supramolecular structures (whether on or off the mineral surface) have the ability to promote the formation of new minerals, such as peptide nanostructures catalyzing the formation of hydroxyapatite, similar to modern-day biomineralization processes. The newly produced mineral surfaces could then catalyze and template the synthesis, polymerization, and/or assembly of the same structures that allowed its own mineralization, leading to a synergistic cyclical positive feedback loop by which primitive self-assemblies could have affected the mineral composition of early Earth, and vice-versa.

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
