# Peer review of "Mineral Surface-Templated Self-Assembling Systems: Case Studies from Nanoscience and Surface Science towards Origins of Life Research"

_life, 2018, doi:10.3390/life8020010_

Round 1

Reviewer 1 Report

In this review, Gillams and Jia emphasize on a ~70-year-old hypothesis of John Desmond Bernal who first suggested that mineral surfaces, particularly clays, could have played critical role leading to the origins of life about 4 billion years ago (Bernal 1949). This hypothesis has gained increasing attention ever since, notably after the discovery of the property of clays as catalysts of (1) electron-transfer reactions (Solomon and Rosser 1965, Solomon 1968), (2) nonenzymatic polymerization of RNA (Ferris and Ertem 1992) and (3) peptides (Rode 1999), and catalysts of (4) lipid membrane self-assembly (Hanczyc et al. 2003).

Additionally, the two major current scenarios of origins of life on Earth also feature the role of minerals. Namely, the role of metal sulfides as catalysts of CO2 reduction by H2 near hydrothermal vents in deep early oceans (Martin et al. 2008), as opposed to a suite of minerals as catalysts of organics’ synthesis and assembly under hydration-dehydration cycles in hydrothermal fields at the early Earth’s surface (Damer and Deamer 2015).

Hence, Gillams and Jia are undoubtedly correct about the topic of their review. Furthermore, the authors aim here to bring information from other fields such as surface chemistry and nanotechnology research in order to widen the field of view for origins of life researchers. Although I believe that surface chemistry has been widely accounted in origins of life research, considering advances in nanotechnology to learn new concepts in origins of life could be novel. Nevertheless, the authors fell short of their goal because the present manuscript came, in general, little weak in the beginning, then showed some potential, but ended up weak again. I believe that this manuscript is still in its preliminary stages and it would necessitate a major revision before considering it for publication.

In detail:

- The title reads “Mineral Surface-Templated Self-Assembling Systems: Geochemical Mechanisms for Achieving Complexity” but I could not find any discussion related to geochemistry—for instance, water/rock interactions and weathering processes. A whole body of modeling and experimental research was omitted. In addition, I would have liked to see more discussion related to “mechanisms”. If this was not the authors’ intent in this review, the title is then misleading.

- Section 2.1 “Mineral Surface-Catalyzed Polymerization of Biomolecules” does not show any novel idea. The authors even reduce the catalytic activity of minerals to a concentration effect that could favor intermolecular interactions. This is wrong because adsorption is not sufficient for non-enzymatic polymerization. See Ferris and Ertem 1992, Miyakawa et al. 2006, where tens of minerals and rocks were tested, but only montmorillonite was catalytic and even not all of montmorillonite were catalytic (Joshi et al. 2009). So “orientational specificity for catalytic selectivity” (Line 45) is much important for polymerization than adsorption. I would have liked to read a discussion about the molecular details for this surface chemistry. If this discussion was not a goal in this manuscript, then this section should be deleted.

- Section 2.2 “Self-Assembly Promoted by Mineral Surfaces” does not include any example showing self-assembly promoted by mineral surfaces. The title then is misleading. I would have liked to see illustrative examples of mineral surfaces catalyzing self-assembly processes. Otherwise, this section should be deleted. Lines 83-90 belong to the introduction.

 - Section 2.3 “small organics” is nicely illustrated and the example of ethanol adsorption on calcite is well described. However, the section ends with a vague proposal:

One may also imagine a scenario where only part of a surface is populated by an organic monolayer, such as on a heterogeneous rock surface with multiple mineral types that have different adsorption propensities of organics [62]. This microenvironment could then result in the simultaneous adsorption and reaction of a diverse set of molecules in different chemical environments, potentially assisting early chemical systems in diversifying and incorporation or coupling of different types of reactions”.

The authors do not explicitly say what could be these rocks or these sets of molecules. And again, adsorption cannot assure catalysis by itself. Orientation on the surface is the key.

- Section 2.4 “Nucleic acids” has a potential. I liked the discussion of nucleic acids conformational changes upon adsorption on minerals as well as the nucleic acid origami idea. But the idea needs more tweaking before adapting it in origins of life research. For instance, what nucleotides/nucleosides/nucleobases could be relevant? Activation source? Mineral repertoire? Amino acids present? Fatty acids present? Etc. These questions should be addressed.

- Section 2.5 “Peptides”. Idem. I like the analogy the authors are making between the current amyloids and amphiphilic peptides on one hand and the primitive self-assembling peptides on the other.

- Section 2.6 “Lipids and vesicles” is to me incomplete. Enumerating the roles of lipids and vesicles in other fields is not sufficient. The authors need to come up with illustrative examples that could be relevant to the prebiotic chemistry/origins of life research.

- “Conclusions”. It is natural to expect a model unifying the role of minerals in origins of life by considering a system in which minerals, nucleic acids, amino acids and lipids (and sugars) are present. But the authors do not acknowledge the pitfalls of such a complex system (figure 6), nor propose a feasible experimental approach to it. Furthermore, a very similar system has been previously proposed (Kaddour and Sahai 2014) and not cited.

Minor issues:

- Line 48: “terrestrial planets, which are simply large rocks [9]”. Terrestrial planets are NOT simply rocks. A meteorite could be a large rock, and at best, one can argue of an asteroid, but not a terrestrial planet. Terrestrial planets are entities with usually a core, a mantle, an atmosphere, surface features such as mountains and canyons, tectonic activities, weather cycles, etc. –they cannot be simply reduced to large rocks.

- Line 49: Reference 10 does not fit the context. Ebisuzaki and Maruyama 2017 brilliantly propose a new model to explain the origins of life. But they did not explain the occurrence of mineral water interface in the universe. I would cite instead a recent review by Westall, F. & Brack, A. Space Sci Rev (2018) 214: 50. https://doi.org/10.1007/s11214-018-0476-7  

- Line 181 is figure 2 not, figure 1.

- Figure 3. I can’t understand it. The figure might be cropped.

- Figure 4: is not cited in the text. The difference in the rates is not explained.

- Figure 6: Add figure keys 

- Line 303, dramatically … Earth’s mineralogy. Missing a word

- Ref. 1 missing title

- Ref. 48 missing year

- Etc.

References:

Bernal, J. D. (1949). "The physical basis of life." Proceedings of the Physical Society. Section A 62(9): 537.

Damer, B. and D. Deamer (2015). "Coupled phases and combinatorial selection in fluctuating hydrothermal pools: A scenario to guide experimental approaches to the origin of cellular life." Life 5(1): 872-887.

Ferris, J. P. and G. Ertem (1992). "Oligomerization of ribonucleotides on montmorillonite: reaction of the 5'-phosphorimidazolide of adenosine." Science 257(5075): 1387-1389.

Ferris, J. P. and G. Ertem (1992). "Oligomerization reactions of ribonucleotides: the reaction of the 5′-phosphorimidazolide of nucleosides on montmorillonite and other minerals." Origins of Life and Evolution of the Biosphere 22(6): 369-381.

Hanczyc, M. M., S. M. Fujikawa and J. W. Szostak (2003). "Experimental models of primitive cellular compartments: encapsulation, growth, and division." Science 302(5645): 618-622.

Joshi, P. C., M. F. Aldersley, J. W. Delano and J. P. Ferris (2009). "Mechanism of montmorillonite catalysis in the formation of RNA oligomers." Journal of the American Chemical Society 131(37): 13369-13374.

Kaddour, H. and N. Sahai (2014). "Synergism and Mutualism in Non-Enzymatic RNA Polymerization." Life 4(4): 598.

Martin, W., J. Baross, D. Kelley and M. J. Russell (2008). "Hydrothermal vents and the origin of life." Nature Reviews Microbiology 6(11): 805.

Miyakawa, S., P. C. Joshi, M. J. Gaffey, E. Gonzalez-Toril, C. Hyland, T. Ross, K. Rybij and J. P. Ferris (2006). "Studies in the mineral and salt-catalyzed formation of RNA oligomers." Origins of Life and Evolution of Biospheres 36(4): 343.

Rode, B. M. (1999). "Peptides and the origin of life1." Peptides 20(6): 773-786.

Solomon, D. (1968). "Clay minerals as electron acceptors and/or electron donors in organic reactions." Clays and Clay Minerals 16(31).

Solomon, D. and M. Rosser (1965). "Reactions catalyzed by minerals. Part I. Polymerization of styrene." Journal of Applied polymer science 9(4): 1261-1271.

Westall, F. and A. Brack (2018). "The importance of water for life." Space Science Reviews 214(2): 50.

Author Response

Reviewer 1

In this review, Gillams and Jia emphasize on a ~70-year-old hypothesis of John Desmond Bernal who first suggested that mineral surfaces, particularly clays, could have played critical role leading to the origins of life about 4 billion years ago (Bernal 1949). This hypothesis has gained increasing attention ever since, notably after the discovery of the property of clays as catalysts of (1) electron-transfer reactions (Solomon and Rosser 1965, Solomon 1968), (2) nonenzymatic polymerization of RNA (Ferris and Ertem 1992) and (3) peptides (Rode 1999), and catalysts of (4) lipid membrane self-assembly (Hanczyc et al. 2003).

Additionally, the two major current scenarios of origins of life on Earth also feature the role of minerals. Namely, the role of metal sulfides as catalysts of CO2 reduction by H2 near hydrothermal vents in deep early oceans (Martin et al. 2008), as opposed to a suite of minerals as catalysts of organics’ synthesis and assembly under hydration-dehydration cycles in hydrothermal fields at the early Earth’s surface (Damer and Deamer 2015).

Hence, Gillams and Jia are undoubtedly correct about the topic of their review. Furthermore, the authors aim here to bring information from other fields such as surface chemistry and nanotechnology research in order to widen the field of view for origins of life researchers. Although I believe that surface chemistry has been widely accounted in origins of life research, considering advances in nanotechnology to learn new concepts in origins of life could be novel. Nevertheless, the authors fell short of their goal because the present manuscript came, in general, little weak in the beginning, then showed some potential, but ended up weak again. I believe that this manuscript is still in its preliminary stages and it would necessitate a major revision before considering it for publication.

We thank the reviewer for their very detailed and constructive critical comments about the manuscript and their agreement with our assertion that mineral surfaces were essential in the initial emergence of life on Earth. Our aim was indeed to emphasize the importance of novel research in self-assembling systems from the nanoscience and surface science fields in origins of life and prebiotic chemistry research, specifically by using examples of mineral surface-templated self-assembling systems as a way to show the relevance of such systems on the prebiotic earth. However, perhaps the title and focus of our review were somewhat unclear and disjointed. Thus, we have made major changes to much of the manuscript, including reorganizing some of the sections, changing the title so that less focus is placed on specific geochemical or chemical mechanisms. We have restructured the document such that it is based around case studies, and focuses more on how to tie in the nanoscience/surface science examples directly with origins of life work. We believe that the manuscript has improved significantly with the suggestions of the reviewer, and hope that the updated manuscript is up to their standards for publication in Life.

In detail:

- The title reads “Mineral Surface-Templated Self-Assembling Systems: Geochemical Mechanisms for Achieving Complexity” but I could not find any discussion related to geochemistry—for instance, water/rock interactions and weathering processes. A whole body of modeling and experimental research was omitted. In addition, I would have liked to see more discussion related to “mechanisms”. If this was not the authors’ intent in this review, the title is then misleading.

We thank the reviewer for this helpful comment. Indeed, it was not our intent to go into depth into geochemical processes in this review. Rather, as the reviewer has pointed out, the goal was to utilize examples from supramolecular self-assembling systems studied in surface science and nanotechnology, each of which is dependent on a mineral surface-templated assembly process, to illustrate the relevance of such systems in achieving more complexity in prebiotic chemistry and origins of life studies. Perhaps the word “mechanism” may have been ambiguous, as our intent was to present the self-assembly process itself as the mechanism (not necessarily a specific chemical mechanism, but rather a more general causal mechanism) by which a system could achieve more complexity, whether structurally or functionally. However, “mechanism” is understood with a different connotation in different fields of research, and thus perhaps the reviewer was (and importantly, if used ambiguously, future readers as well would be) misled into thinking that there would be more discussion in the geochemical or chemical mechanisms by which the mineral surfaces are specifically catalyzing or templating the self-assembly process. While we do focus in some instances on chemical mechanisms, some of the examples that we presented do not have a well-understood chemical mechanism. And thus, in order to avoid further ambiguity and given that we hope to provide an accessible bridge between fields, we have removed the words “geochemical” and “mechanism” from the title and have instead focused on the novel aspects of the review, namely the integration of novel ideas from nanoscience and surface science, while at the same time modifying our abstract. Our new title is “Mineral Surface-Templated Self-Assembling Systems: Case Studies from Nanoscience and Surface Science towards Origins of Life Research”. We hope that this title is less misleading and ambiguous.

- Section 2.1 “Mineral Surface-Catalyzed Polymerization of Biomolecules” does not show any novel idea. The authors even reduce the catalytic activity of minerals to a concentration effect that could favor intermolecular interactions. This is wrong because adsorption is not sufficient for non-enzymatic polymerization. See Ferris and Ertem 1992, Miyakawa et al. 2006, where tens of minerals and rocks were tested, but only montmorillonite was catalytic and even not all of montmorillonite were catalytic (Joshi et al. 2009). So “orientational specificity for catalytic selectivity” (Line 45) is much important for polymerization than adsorption. I would have liked to read a discussion about the molecular details for this surface chemistry. If this discussion was not a goal in this manuscript, then this section should be deleted.

We have deleted much of this section and incorporated the remaining important points into the introduction.

- Section 2.2 “Self-Assembly Promoted by Mineral Surfaces” does not include any example showing self-assembly promoted by mineral surfaces. The title then is misleading. I would have liked to see illustrative examples of mineral surfaces catalyzing self-assembly processes. Otherwise, this section should be deleted. Lines 83-90 belong to the introduction.

We have incorporated much of this section into the introduction.

 - Section 2.3 “small organics” is nicely illustrated and the example of ethanol adsorption on calcite is well described. However, the section ends with a vague proposal:

“One may also imagine a scenario where only part of a surface is populated by an organic monolayer, such as on a heterogeneous rock surface with multiple mineral types that have different adsorption propensities of organics [62]. This microenvironment could then result in the simultaneous adsorption and reaction of a diverse set of molecules in different chemical environments, potentially assisting early chemical systems in diversifying and incorporation or coupling of different types of reactions”.

The authors do not explicitly say what could be these rocks or these sets of molecules. And again, adsorption cannot assure catalysis by itself. Orientation on the surface is the key.

We thank the reviewer for this comment. We now explain this in further detail (now section 2.1) and add a paragraph suggesting heterogeneous rock surfaces, such as shales, could have facilitated this type of microenvironment. Shales contain multiple mineral types, and different small organics have been shown to have strong binding or adsorption to certain minerals. Different clays have also shown different propensities for adsorption of various organics, including small polar organics like formamide or urea strongly adsorbing onto kaolinites and various simple aromatic compounds like benzene and phenol interacting with Cu-Montmorillonite (Lagaly, G.; Barrer, R. M.; Goulding, K. Clay-Organic Interactions. Philosophical Transactions of the Royal Society A: Mathematical, Physical and Engineering Sciences 1984, 311, 315–332.), while amino acids have been shown to have specific preference for binding to certain minerals, such as aspartic acid preferentially adsorbing onto calcites and glycine, serine, isoleucine, leucine and arginine preferentially adsorbing onto quartz (W. Carter, P. Adsorption of amino acid-containing organic matter by calcite and quartz. Geochim. Cosmochim. Acta 1978, 42, 1239–1242.). Such isolated organic environments could have resulted in specific reactions in the respective adsorbed regions. We also suggest that the close proximity of multiple different types of reactions could have assisted in the incorporation or coupling of different types of reactions.

- Section 2.4 “Nucleic acids” has a potential. I liked the discussion of nucleic acids conformational changes upon adsorption on minerals as well as the nucleic acid origami idea. But the idea needs more tweaking before adapting it in origins of life research. For instance, what nucleotides/nucleosides/nucleobases could be relevant? Activation source? Mineral repertoire? Amino acids present? Fatty acids present? Etc. These questions should be addressed.

We thank the reviewer for these comments and have tried to tie our case studies in more with origins of life studies. We have expanded upon the section (now 2.2) where we originally discuss DNA conformational changes upon mineral binding. We now add a few sentences where we speculate about the possibility of and interest in studying in alternative forms of RNA induced by mineral surface adsorption (which would have been more relevant to origins of life research than DNA, although very little work has been done on RNA conformational changes induced by mineral binding). The introduction of novel RNA structures (for example, Z-form RNA composed of CG-repeats, both of which may have been more abundant on the early earth due to their lower error propensity and faster polymerization rate in non-enzymatic replicating RNA systems compared to A and U, or RNA containing Z-like steps) in an early genetic replicating system may have allowed such a system to explore additional sequence space for novel functions.

We have also elaborated much more on nucleic acid structures on adsorbed mica surfaces induced by various divalent cation solutions. We have included a brief description of the potential reasons of the differential structure formation caused by cobalt cations versus magnesium ions and also other possible divalent systems that could also result in similar assemblies on mica surfaces. In particular, we propose that it would be of interest to the research community to study the effect of ferrous iron on DNA adsorption and assembly on mica surfaces due to its similar physical properties (namely ionic radius) to cobalt and due to its abundance on early earth oceans, as compared to the likely much lower concentration of cobalt present on the early Earth. Finally, we elaborate a bit more on the RNA origami example.

- Section 2.5 “Peptides”. Idem. I like the analogy the authors are making between the current amyloids and amphiphilic peptides on one hand and the primitive self-assembling peptides on the other.

We thank the reviewer for this comment. We now add in a bit more detail about self-assembled nanofibers and their analogy to peptide dendrimer enzymes, as well as specific structures and functions of amphiphilic peptides and peptide amphiphiles that would be relevant to prebiotic chemistry research (now section 2.3).

- Section 2.6 “Lipids and vesicles” is to me incomplete. Enumerating the roles of lipids and vesicles in other fields is not sufficient. The authors need to come up with illustrative examples that could be relevant to the prebiotic chemistry/origins of life research.

We agree with the reviewer that simply reviewing research without presenting novel ideas relevant to origins of life research is not sufficient. Because of this, and due to the lack of concrete examples that we have found in this topic as well as the increased length due to additions in other sections, we have moved some of the essential information regarding mineral-surface templated vesicle formation into the introduction section, and deleted much of the rest of the section so that it does not detract from the purpose of the rest of the case studies.

- “Conclusions”. It is natural to expect a model unifying the role of minerals in origins of life by considering a system in which minerals, nucleic acids, amino acids and lipids (and sugars) are present. But the authors do not acknowledge the pitfalls of such a complex system (figure 6), nor propose a feasible experimental approach to it. Furthermore, a very similar system has been previously proposed (Kaddour and Sahai 2014) and not cited.

We have modified our proposed system slightly to focus more on how a self-assembly/biomineralization cycle could have arisen and have also moved this entire subsection into the previous large section. We now believe our proposed system is somewhat different from Kaddour and Sahai’s proposed system. Kaddour and Sahai’s system mainly describes how peptides and RNA could have arisen synergistically on the early Earth, potentially with the help of encapsulating vesicles. We have also modified and simplified the figure (now figure 5) by removing some shape-forms, modifying the shapes, and now include a figure legend explaining the components in greater detail. We hope that this has cleared up some of the confusion surrounding this figure and that it is clear that this figure describes a proposed self-assembly/mineralization cycle rather than a mineral-based concerted origins of life system. Our figure specifically does not identify specific types of molecules/polymers, but rather focuses on the nature of the processes that can lead to enhanced complexity.

Minor issues:

- Line 48: “terrestrial planets, which are simply large rocks [9]”. Terrestrial planets are NOT simply rocks. A meteorite could be a large rock, and at best, one can argue of an asteroid, but not a terrestrial planet. Terrestrial planets are entities with usually a core, a mantle, an atmosphere, surface features such as mountains and canyons, tectonic activities, weather cycles, etc. –they cannot be simply reduced to large rocks.

We understand that perhaps we made a gross oversimplification of terrestrial planets due to the way that we wrote this statement. Our intent was simply to state that mineral surfaces were present on terrestrial planets and that if mineral surfaces did play a large role in the initial emergence of life on our own terrestrial planet, then understanding these processes would likely give us greater insights into the potential emergence of life on any terrestrial planet. Thus, we have changed the wording of this statement.

- Line 49: Reference 10 does not fit the context. Ebisuzaki and Maruyama 2017 brilliantly propose a new model to explain the origins of life. But they did not explain the occurrence of mineral water interface in the universe. I would cite instead a recent review by Westall, F. & Brack, A. Space Sci Rev (2018) 214: 50. https://doi.org/10.1007/s11214-018-0476-7

The original Ebisuzaki and Maruyama citation was included to emphasize the importance of aqueous-mineral interactions in prebiotic chemistry, as their model shows many examples by which aqueous-mineral interactions were key to the development of an early living system on Earth. Perhaps our original statement was not well-conveyed as our intention was to infer that water-mineral interactions are likely to have been important in the origin of life on any terrestrial body due to the importance of water-mineral interactions in the origin of life on Earth. This assumes that life off of Earth should be similar in form as that on Earth and that the mechanism of the emergence of life extraterrestrially should be similar of that on Earth. In any case, we thank the reviewer for pointing out this reference, and we have rewritten this section and included the suggested reference such that hopefully it is more clear in presenting our point.

- Line 181 is figure 2 not, figure 1.

We have changed the figure caption.

- Figure 3. I can’t understand it. The figure might be cropped.

Figure 3 shows the self-assembled fibrils from a peptide self-assembling system on mica. Figures A-E show various images of the fibrils at 12-16 hours (it is not a timecourse), while Figure F shows an image of the fibrils at 6 days. Longer incubation times appear to induce longer fibrils compared to shorter incubation times. Figure 3 was included to provide a tangible example of a peptide system that is able to self-assemble into specific nanostructures on a mineral surface. We are unsure as to how the figure may have been cropped incorrectly, as both the figure that we included in the submitted word document, as well as the figure available for download show the whole figure. We would appreciate if the reviewer is able to describe in more detail what may be wrong with the figure so that if it is a technical issue, then we can try to remedy it as quickly as possible. Nevertheless, we have include slightly more description in the figure caption.

- Figure 4: is not cited in the text. The difference in the rates is not explained.

As part of the restructuring described above Figure 4 has now been removed. It was an oversight not to cite it in the original text, but in the context of the new document format, the figure is no longer required.

- Figure 6: Add figure keys

We have modified and simplified the figure (now Figure 5) and now include a figure legend explaining the components in greater detail. Given that the purpose of the figure is to highlight underlying processes rather than specific types of molecules, the legend uses intentionally broad terminology to describe the chemical entities in play and focuses on the interaction between these entities.

- Line 303, dramatically … Earth’s mineralogy. Missing a word

We have fixed this sentence.

- Ref. 1 missing title

We believe that the reviewer may be mistaken. In Ref. 1, “Origens da vida” is the title. Estud. av. (Estudos Avançados) is the journal. (dx.doi.org/10.1590/S0103-40142007000100022)

- Ref. 48 missing year

We have added this information

- Etc.

As part of the restructuring of the document a number of minor errors have been rectified, so we hope these have been ironed out.

References:

Bernal, J. D. (1949). "The physical basis of life." Proceedings of the Physical Society. Section A 62(9): 537.

Damer, B. and D. Deamer (2015). "Coupled phases and combinatorial selection in fluctuating hydrothermal pools: A scenario to guide experimental approaches to the origin of cellular life." Life 5(1): 872-887.

Ferris, J. P. and G. Ertem (1992). "Oligomerization of ribonucleotides on montmorillonite: reaction of the 5'-phosphorimidazolide of adenosine." Science 257(5075): 1387-1389.

Ferris, J. P. and G. Ertem (1992). "Oligomerization reactions of ribonucleotides: the reaction of the 5′-phosphorimidazolide of nucleosides on montmorillonite and other minerals." Origins of Life and Evolution of the Biosphere 22(6): 369-381.

Hanczyc, M. M., S. M. Fujikawa and J. W. Szostak (2003). "Experimental models of primitive cellular compartments: encapsulation, growth, and division." Science 302(5645): 618-622.

Joshi, P. C., M. F. Aldersley, J. W. Delano and J. P. Ferris (2009). "Mechanism of montmorillonite catalysis in the formation of RNA oligomers." Journal of the American Chemical Society 131(37): 13369-13374.

Kaddour, H. and N. Sahai (2014). "Synergism and Mutualism in Non-Enzymatic RNA Polymerization." Life 4(4): 598.

Martin, W., J. Baross, D. Kelley and M. J. Russell (2008). "Hydrothermal vents and the origin of life." Nature Reviews Microbiology 6(11): 805.

Miyakawa, S., P. C. Joshi, M. J. Gaffey, E. Gonzalez-Toril, C. Hyland, T. Ross, K. Rybij and J. P. Ferris (2006). "Studies in the mineral and salt-catalyzed formation of RNA oligomers." Origins of Life and Evolution of Biospheres 36(4): 343.

Rode, B. M. (1999). "Peptides and the origin of life1." Peptides 20(6): 773-786.

Solomon, D. (1968). "Clay minerals as electron acceptors and/or electron donors in organic reactions." Clays and Clay Minerals 16(31).

Solomon, D. and M. Rosser (1965). "Reactions catalyzed by minerals. Part I. Polymerization of styrene." Journal of Applied polymer science 9(4): 1261-1271.

Westall, F. and A. Brack (2018). "The importance of water for life." Space Science Reviews 214(2): 50.

Reviewer 2 Report

The authors Richard Gillams and Tony Jia present a review on the possible role of mineral surfaces templating molecular self-assembly during the origin and early evolution of life. While this topic has been subject of previous reviews in the past, the authors whish to provide new context from other fields (nanotechnology, surface science) in order to help motivate new ideas and research. While this review is not exhaustive, it does provide some ideas that may be of interest to the field of origins of life research. A major issue I see with this article is that many of the concepts are not fully developed, or presented in a way that makes the stated idea clear. I have listed these below. I suggest accepting the article after revisions

Corrections/points of concern

1. Line 2, Title. Very few geochemical mechanisms are being described in this review and when they are they are not described in much detail. It would be more truthful to make the title reference the authors’ new approach to this area of origins research from perspectives in different fields (nanotech, surface science). Also, because they focus only on a few examples (such as DNA helix transitions mediated by minerals) they might want to make it clear in the title that only a ‘few examples” are discussed.

2. Line 49. This sentence contains several ideas, and a clear point is not present.

“ As mineral surfaces are likely to be similar in structure universally, they could potentially provide ubiquitous interfacial catalysis that may be important in the origin and evolution of life on any rocky planet.”

Do you mean that because they are similar to the minerals on Earth than studying mineral surfaces in the laboratory could provide insight into the origin and evolution of life elsewhere?

-        

2. Line 57, remove ‘to’

3. Line 59, when referencing borates in nucleic acid assembly please cite Benner’s original Science paper on the topic (which was done in water) in addition to the word by Saladino and Di Mauro in formamide. In this section I recommend citing Matt Pasek who has worked most recently on phosphorylation of nucleosides to make nucleotides and other organics by phosphate minerals (schreibersite, ect.).

4. Line 65.  The claim that absorption to minerals will enhance the kinetics (suggesting rate constants and not mass action effects) of a reaction is not something that is apparent, and the reference that is given discusses rate enhancement on cell surfaces, which are fluidic (liquid crystalline) and nothing like a static crystalline surface of a mineral. This needs to be removed or a new reference should be found (such as one that describes true catalysis on minerals – i.e., rate enhancements).

4. Line 136, I am not sure of an enzyme that unwinds a double helix during translation since mRNA is single stranded and tRNA does not need to unwind, furthermore this line was eluding to DNA, which is not used in translation.

5. Lines 150-166, This paragraph describes how minerals influence the possible double helical structures of DNA (A, B, C, Z), and why this might have been important during the origin of life.

First, DNA is not well accepted to have been around before the advent of encapsulation and enzymes (JD Sutherland might have some proposals in articles but they are not well accepted or established experimentally). For this reason a discussion of how this relates to RNA (or pre-RNA) would be prudent, as RNA also can transition to different forms upon binding of small molecules or in different solution environments. While this might have been omitted because little has been done with RNA, I think it needs to be noted that there is no established theory that DNA preceded cellular compartmentalization and RNA might very well undergo similar transitions – which would then be interesting to some researchers in the field.

            Second, starting on line 156 “After desorption from the kaolinite, double-stranded DNA then changes to the seldom-observed C-form helical conformation.” This is interesting but I dont understand how desorption from a mineral can capture C-form structures, these forms are dynamic and dependent on hydration and binding of ions to the major and minor grooves, so once it is desorbed it should not have a ‘memory’ of the mineral and will return to the most energetically favorable form in solution (likely B-form), potentially the salinity of these solutions drove it to C-form or some other binding event but not ‘memory’ of the mineral. Please clarify why C-form is maintained in solution after desorption from a mineral.

6. Line 168, again please try to provide some chemical rational when you write that some things work and some things don’t. It is known that divalent cations (Mg and Co) adsorb DNA to mica and monovalents (Na) do not. Why are different structures formed? Also, how much cobalt (II) really might have been around? Does this work with iron (II), which would have been ubiquitous?

7. The conclusion is not well organized. The subsection 3.1 (there is no 3.2) would, in my opinion, fit better in the discussion section as it presents new ideas from fields that were not discussed prior. I suggest moving this discussion up.

8. line 303, “dramatically [verb] Earth’s…”

Author Response

Reviewer 2

The authors Richard Gillams and Tony Jia present a review on the possible role of mineral surfaces templating molecular self-assembly during the origin and early evolution of life. While this topic has been subject of previous reviews in the past, the authors whish to provide new context from other fields (nanotechnology, surface science) in order to help motivate new ideas and research. While this review is not exhaustive, it does provide some ideas that may be of interest to the field of origins of life research. A major issue I see with this article is that many of the concepts are not fully developed, or presented in a way that makes the stated idea clear. I have listed these below. I suggest accepting the article after revisions

We thank the reviewer for carefully reviewing our article and providing constructive critical comments. We have attempted to modify the manuscript according to their suggestions. Specifically, we have reorganized some of the sections as to more carefully delineate specific ideas from each other, while also briefly expounding a bit more on some of the case studies that we have presented. We hope that the reviewer finds our updated manuscript worthy of publication in Life.

Corrections/points of concern

1. Line 2, Title. Very few geochemical mechanisms are being described in this review and when they are they are not described in much detail. It would be more truthful to make the title reference the authors’ new approach to this area of origins research from perspectives in different fields (nanotech, surface science). Also, because they focus only on a few examples (such as DNA helix transitions mediated by minerals) they might want to make it clear in the title that only a ‘few examples” are discussed.

We thank the reviewer for this comment. Indeed, it was not our intent to go into depth into geochemical processes in this review. Rather, as the reviewer has pointed out, the goal was to utilize examples from supramolecular self-assembling systems studied in surface science and nanotechnology, each of which is dependent on a mineral surface-templated assembly process, to illustrate the relevance of such systems in achieving more complexity in prebiotic chemistry and origins of life studies. Perhaps the word “mechanism” may have been ambiguous, as our intent was to present the self-assembly process itself as the mechanism (not necessarily a specific chemical mechanism, but rather a more general causal mechanism) by which a system could achieve more complexity, whether structurally or functionally. However, “mechanism” is understood with a different connotation in different fields of research, and thus perhaps the reviewer ultimately was (and importantly, if used ambiguously, future readers as well would be) misled into thinking that there would be more discussion in the geochemical or chemical mechanisms by which the mineral surfaces are specifically catalyzing or templating the self-assembly process. While we do focus in some instances on chemical mechanisms, some of the examples that we presented do not have a well-understood chemical mechanism. And thus, in order to avoid further ambiguity and given that we hope to provide an accessible bridge between fields, we have removed the words “geochemical” and “mechanism” from the title and have instead focused on the novel aspects of the review, namely the integration of novel ideas from nanoscience and surface science, while at the same time modifying our abstract. Our new title is “Mineral Surface-Templated Self-Assembling Systems: Case Studies from Nanoscience and Surface Science towards Origins of Life Research”. We hope that this title is less misleading and ambiguous, and also emphasizes that only a few examples will be discussed.

2. Line 49. This sentence contains several ideas, and a clear point is not present.

“ As mineral surfaces are likely to be similar in structure universally, they could potentially provide ubiquitous interfacial catalysis that may be important in the origin and evolution of life on any rocky planet.”

Do you mean that because they are similar to the minerals on Earth than studying mineral surfaces in the laboratory could provide insight into the origin and evolution of life elsewhere?

This was indeed our intention, and we have made the text clearer to reflect this idea.

2. Line 57, remove ‘to’

We have amended this error.

3. Line 59, when referencing borates in nucleic acid assembly please cite Benner’s original Science paper on the topic (which was done in water) in addition to the word by Saladino and Di Mauro in formamide. In this section I recommend citing Matt Pasek who has worked most recently on phosphorylation of nucleosides to make nucleotides and other organics by phosphate minerals (schreibersite, ect.).

We thank the reviewer for these suggestions, and have incorporated these references into the text.

4. Line 65.  The claim that absorption to minerals will enhance the kinetics (suggesting rate constants and not mass action effects) of a reaction is not something that is apparent, and the reference that is given discusses rate enhancement on cell surfaces, which are fluidic (liquid crystalline) and nothing like a static crystalline surface of a mineral. This needs to be removed or a new reference should be found (such as one that describes true catalysis on minerals – i.e., rate enhancements).

We have removed this section from the introduction, as this is beyond the scope of our review.

4. Line 136, I am not sure of an enzyme that unwinds a double helix during translation since mRNA is single stranded and tRNA does not need to unwind, furthermore this line was eluding to DNA, which is not used in translation.

This was our oversight and we have removed the mention of “translation”.

5. Lines 150-166, This paragraph describes how minerals influence the possible double helical structures of DNA (A, B, C, Z), and why this might have been important during the origin of life.

First, DNA is not well accepted to have been around before the advent of encapsulation and enzymes (JD Sutherland might have some proposals in articles but they are not well accepted or established experimentally). For this reason a discussion of how this relates to RNA (or pre-RNA) would be prudent, as RNA also can transition to different forms upon binding of small molecules or in different solution environments. While this might have been omitted because little has been done with RNA, I think it needs to be noted that there is no established theory that DNA preceded cellular compartmentalization and RNA might very well undergo similar transitions – which would then be interesting to some researchers in the field.

We thank the reviewer for this suggestion. We have included a statement which explains that although RNA would likely have been more relevant to prebiotic chemistry than DNA, because there haven’t been many studies related to studying various RNA conformations upon mineral adsorption/desorption, we used DNA as the case studies. One could imagine that similar environments could have induced RNA to undergo similar types of transitions.

Second, starting on line 156 “After desorption from the kaolinite, double-stranded DNA then changes to the seldom-observed C-form helical conformation.” This is interesting but I dont understand how desorption from a mineral can capture C-form structures, these forms are dynamic and dependent on hydration and binding of ions to the major and minor grooves, so once it is desorbed it should not have a ‘memory’ of the mineral and will return to the most energetically favorable form in solution (likely B-form), potentially the salinity of these solutions drove it to C-form or some other binding event but not ‘memory’ of the mineral. Please clarify why C-form is maintained in solution after desorption from a mineral.

We thank the reviewer for pointing this out. Upon further examination, the authors of the paper we referenced provided no explanation into the mechanism of such a shift. It appears that they performed CD spectroscopy on desorbed DNA, and inferred that anything that was desorbed from the mineral was exactly what was originally on the mineral, which may be incorrect as the reviewer has correctly pointed out that conformation is dependent on solution conditions. (although there is a chance that it has reached a kinetic trap which would perhaps not allow such a structure to transition back into the original apo state conformation). Because of the unclear nature of this part of the original study, we believe that it is most prudent to remove the sections describing C-form DNA from our manuscript and focus on the parts of their study which should still be sound, which is the detection of  Z-form DNA via FTIR upon kaolinite binding.

6. Line 168, again please try to provide some chemical rational when you write that some things work and some things don’t. It is known that divalent cations (Mg and Co) adsorb DNA to mica and monovalents (Na) do not. Why are different structures formed? Also, how much cobalt (II) really might have been around? Does this work with iron (II), which would have been ubiquitous?

We have elaborated much more on nucleic acid structures on adsorbed mica surfaces induced by various divalent cation solutions. We have included a brief description of the potential reasons of the differential structure formation caused by cobalt cations versus magnesium cations and also other possible divalent systems that could also result in similar assemblies on mica surfaces. In particular, we propose that it would be of interest to the research community to study the effect of ferrous iron on DNA adsorption and assembly on mica surfaces due to its similar physical properties (namely ionic radius) to cobalt and due to its abundance on early earth oceans, as compared to the likely much lower concentration of cobalt present on the early Earth.

7. The conclusion is not well organized. The subsection 3.1 (there is no 3.2) would, in my opinion, fit better in the discussion section as it presents new ideas from fields that were not discussed prior. I suggest moving this discussion up.

We have moved the portion of subsection 3.1 where we discuss biomineralization into a new section in section 2 (2.4 Biomineralization Processes). In the prospective section (in lieu of a conclusion), we expand a bit more regarding the scheme figure (Figure 5), which is a summary figure and possible model (composed entirely of abstract molecules) of how a mineral-self-assembly templating cycle could have arisen. Finally, we have also written more about linking together novel ideas from surface science and nanotechnology into origins of life research.

8. line 303, “dramatically [verb] Earth’s…”

We have fixed this sentence.

Round 2

Reviewer 1 Report

The authors have dissipated my reservation about their original manuscript by overhauling the text. I thank them in my turn for their point-by-point reply. Furthermore, I liked the new addition of the biomineralization section as well as the prospective sections, as opposed to the conclusions section in the previous version. In conclusion, the manuscript is now to me clear, consistent and well structured. It would be a pleasure to read in the community of origins of life research.